# Healthcare Policies to Eliminate Neglected Tropical Diseases (NTDs) in India: A Roadmap

**DOI:** 10.3390/ijerph20196842

**Published:** 2023-09-27

**Authors:** Ajay Chandra, S. D. Sreeganga, Nibedita Rath, Arkalgud Ramaprasad

**Affiliations:** 1School of Arts, Humanities and Social Sciences, Chanakya University, Bengaluru 562110, India; ajay.c@chanakyauniversity.edu.in; 2Jindal School of Government and Public Policy, O.P. Jindal Global University, Sonipat 131001, India; ssd@jgu.edu.in; 3Open Source Pharma Foundation, National Institute of Advanced Studies, Bengaluru 560012, India; nibedita.rath@ospfound.org; 4Information and Decision Sciences, University of Illinois at Chicago, Chicago, IL 60605, USA

**Keywords:** neglected diseases, healthcare policy, India, ontology, framework, roadmap

## Abstract

The need for systemic healthcare policies to systematically eliminate NTDs globally and in India has been stressed for more than two decades. Yet, the present policies and the research on them do not meet the need. We present an ontological framework, a research roadmap, and a policy brief to address the gap. The ontology clearly, concisely, and comprehensively represents the combinations of diseases, the objectives regarding the diseases, the entities to address them, the outcomes sought, and the potential policy instruments to invoke. The paper explicates the state of the-policies and state of the research on policies to eliminate NTDs in India. It highlights the significant gaps in the diseases covered, balance in the objectives, comprehensiveness of policies, portfolio of outcomes, and involvement of entities. Last, it presents a set of systemic policies congruent with the ontology to systematically address the gaps. The recommendations are aligned with the present research, policies, practices, and recommendations in India and of the WHO, UN agencies, and other similar bodies. The approach can be generalized to provide roadmaps for other countries facing a similar challenge and for other diseases of similar complexity. The roadmaps, with continuous feedback and learning, can help navigate the challenge efficiently and effectively.

## 1. Introduction

Nearly one billion of the world’s population endure neglected tropical diseases (NTDs), which are referred to as “neglected” because they are given little attention by policymakers, have low or no priority within health strategies, have inadequate research funding, given limited resource allocation, and lead to few interventions. These diseases continue to cause significant morbidity and mortality in the developing world. The need for systemic healthcare policies to systematically eliminate NTDs globally and in India has been stressed for more than two decades. A similar need has been advocated for the BRICS (Brazil, Russia, India, China, and South Africa) countries [1,2,3]. Yet, the present policies and the research on them do not meet the need.

We present an ontological framework to help visualize the pathways to address the challenge. We use the framework to describe the gaps in policies, practices, and research on the policies to address the same. Based on the gap analysis we propose a roadmap, based on the ontology, to formulate policies systemically and systematically. 

Two decades ago, Trouiller et al. [4] highlighted the “lack of effective, safe, and affordable pharmaceuticals to control infectious diseases that cause high mortality and morbidity among poor people in the developing world.” Chaudhuri [5] reiterated it about a decade later for India. Effective policies for drug development to eliminate NTDs have not yet evolved globally and in India. Some attempts have been criticized for lack of details [6]. Neither priority setting for [7] nor program integration [8] of policies by a country has been formalized. The search for solutions has continued to be focused on the biomedical sciences to the exclusion of social and ecological sciences and interdisciplinary approaches [9]. Some have also focused on public–private partnerships and international collaborations [5], although not systematically.

The eight-point manifesto Hotez and Pecoul [10] presented in 2010 does not appear to have mobilized effective healthcare policies to eliminate NTDs in India over the past decade. The World Health Assembly and the World Health Organization (WHO) Regional Committee resolutions intended to have a “quick and dramatic impact” [11] do not appear to have been effective. The “audacious goal” [12] is still relevant more than a decade later but unfulfilled. A decade later, it is time to assess the “roadmap for implementation. It represents the next step forward in relieving and, in many cases, finally ending the vast misery caused by these ancient diseases of poverty” [13]. There was a similar call for “road map that delivers a clear vision towards zero new infection by designing low-cost prevention and control strategies” [14] from Tungiasis. The call articulated the need “to develop culturally appropriate communication techniques and a globally scalable collaboration amongst the stakeholders of endemic countries”.

Despite marginal improvements in the drug and vaccine for NTDs, there is a persistent insufficiency in drug and vaccine development for NTDs [15]. The need for a coordinated mechanisms in terms of disease surveillance, prevention, treatment, control, and elimination of NTDs in India remains unmet. A strategic research agenda for engaging in basic, operational, and translational research is needed to inform effective research and development [16].

In 2014, Brazil, Russia, India, China, and South Africa (BRICS) sought to build “on their individual and collective mechanisms for shaping policy and promoting cooperation… [and] shape the policy agenda, increasing political commitment, mobilizing resources and implementing policies that support control and elimination of NTDs at the international level.” [2]. Further, it is important to note that disease burden and domestic policies in BRICS exert primary influence on the trajectory of NTDs research [1].

If India envisages to eliminate NTDs, firstly, it should develop a coherent strategic direction towards research in diseases for which it has signed up for elimination. Secondly, a concerted research strategy is needed to assist the elimination efforts. A more robust framework for bridging the gap between evidence to policy and implementation should become operative [17]. 

The elimination of NTDs could fail if the narrow reductionist approaches for NTDs disease controls are adopted. NTDs are a complex set of diseases which need to be addressed with policy and governance mechanisms that are sensitive to understand the socio-ecological context of NTDs and account for the underlying component of parasite transmission for each class of NTDs [18]. For instance, the shift from biomedical diagnosis approaches to data linkages for automation of surveillance and involvement of public private partnerships should be prioritized for the control and elimination of NTDs [19].

Focusing on the single nation approach could dramatically advance the global health agenda. If the global community focuses on India’s NTDs problem and makes inroads [20], it will substantially reduce the burden of poverty-related NTDs. Further, the need for reframing the issues pertaining to orphan drug issue for achieving sustainable and inclusive health perspectives is timely and critical [21]. In addition, fostering innovations in drugs, diagnostics, and vaccines [22] has to be intricated as part of India’s NTD elimination programs. 

Given India’s NTDs burden, the WHO’s effective roadmap to eliminate India’s NTDs [13] could be indicative for formulation of effective strategies. Applying systems thinking to eliminate NTDs, Glenn et al. [23] propose the following approaches: “(1) clarify the potential for and assess realistic progress towards NTDs elimination, (2) increase support for interventions besides drug delivery, (3) reduce dependency on international donors, (4) create a less insular culture within the global NTDs community, and (5) systemically address the issue of health worker incentives”.

In India, the absence of systemic policies and their systematic implementation has resulted in further neglect of these diseases. The existing health policies in India are selective and siloed; policies such as the National Health Policy, 2017 [24], National Policy on Treatment of Rare Diseases, 2018 [25], Science, Technology, and Innovation Policy, 2020 [26], and Draft National Pharmaceutical Policy, 2017 [27] have failed to integrate specific policies for neglected diseases. Current practices are programmatic and mainly focus on prevention, control, eradication, and elimination mechanisms. In addition, factors like poverty, geographical isolation, stigmatization, paucity of disease burden data, lack of political will, and financial resources have fragmented the approaches in the domain. Thus, there is a need for an integrated platform that can support and ensure systematic and adequate action.

This paper presents an ontological framework for healthcare policies to eliminate NTDs in India to address the needs described earlier. First, we describe the framework. Second, we summarize the state of the policies and state of the research on the policies in India using the framework. Third, we present guidelines for policies and research to address the need. Last, we conclude with a discussion of how the framework and the method can be generalized to other countries that face a similar challenge.

## 2. Materials and Methods—Ontology of Healthcare Policies to Eliminate Neglected Tropical Diseases (NTDs) in India

A sustainable and inclusive healthcare system must be developed to reframe the approaches to tackle NTDs systematically. NTDs denotes a range of infectious, non-infectious, and co-infectious diseases listed in the ontology (Figure 1—disease), based on the taxonomy of Utzinger et al. [28] and papers by Menon et al. [29] Sheshadri et al. [30], Wilairatna et al. [31], and Li and Zhou [32]. The objectives of healthcare policies for NTDs in India should be to identify, prevent, cure, palliate, rehabilitate, educate, eradicate, eliminate, and control (Figure 1—objective) them and their effects. The taxonomy extends the ones used by Sastry et al. [33] for health care policies in India, Dai et al. [34] for China, and Ramaprasad et al. [35] for Australia. ‘Elimination’ is the commonly stated objective and has become almost inextricably associated with NTDs. However, it must be one of the many possible objectives regarding NTDs.

Achieving the objectives would require different constitutional, legislative, economic, regulatory, fiscal/financial, informational, contractual, legal, social, and research and development (R&D) policy mechanisms (Figure 1—policy). The taxonomy is an extension of the one proposed by Lascoumes et al. [36]. These actions must be designed to assure the physical, mental, economic, environmental, social, and societal wellbeing (Figure 1—outcome) of the population. Entities such as (a) academia, (b) public, private, and NGO institutions, (c) personnel consisting of doctors, paramedics, nurse, and staff, (d) communities, (e) family, and (f) individuals (Figure 1—entity) must become the agents for implementing these policies. The last two taxonomies are an aggregation and organization of the outcomes and entities stated in the research and policy documents on NTDs (for example, [7,25,37,38]), and drawn from the authors’ earlier work on healthcare policies [33,34,35]. This logic is encapsulated in the ontology of healthcare policies to eliminate NTDs in India (Figure 1).

The ontology expresses the policy pathways to address the problem; each pathway may be instantiated in many ways. Two illustrative pathways and two possible instantiations of each are given below: Fiscal/financial policy for the cure of infectious helminth diseases by public institutions for societal wellbeing. ○Funding for public health centres for identifying and curing infectious helminth diseases in a community. For example, financial resources of the government at various levels have not been adequate to provide sufficient budget support and personnel dedicated to prevention and control including human deworming in endemic areas [39].○Economic incentives for academia for education on infectious helminth diseases in a community.
R&D policy for identification of co-infectious diseases by academia for physical wellbeing.○Research agenda and grants for academic research on the prevalence of the effects of co-infectious diseases on physical wellbeing.○Research agenda and grants for academic research on the assessment of the treatment of co-infectious diseases on physical wellbeing.


The ontology expresses 10 × 9 × 9 × 11 × 6 = 53,460 such logical pathways with potentially multiple instantiations of each. Each pathway is a concatenation of an element from each column of the ontology, with the adjacent connecting words/phrases. It is a clear, concise, comprehensive visualization of the complexity of the challenge of eliminating NTDs. It can be used like a ‘Google Map’ to systematically assess and formulate a system of policies based on (a) the diseases they address, (b) the objectives regarding the diseases, (c) the entities for addressing the diseases, (d) the outcomes sought, and (e) the policy instruments they invoke. The feasibility and effectiveness of the logical pathways, and consequently the policies, will depend on a variety of budgetary, infrastructural, technical, socio-cultural, and other drivers, norms, and barriers. The dynamics of such forces has been discussed in the context of access to healthcare in Chile [40], during COVID-19 [41], and mental healthcare in Chile [42]. 

### 2.1. Policies on and Practices Regarding NTDs in India

We systematically searched for documents on policies and practices regarding or related to NTDs in India available online from the Ministry of Health and Family Welfare, National Health Authority, and NITI Aayog. The search included those by world agencies such as the World Health Organization and the United Nations Organization. We relied on the expertise of one of the authors, who works in the field, and her expert network to ensure that our search was comprehensive. Subsequently, we collected and collated the documents in a Zotero database. 

### 2.2. Research on NTDs Policies in India

We searched for the literature using the research indexing service Elsevier’s Scopus. The search term TITLE-ABS-KEY ((neglected W/5 disease AND (policy OR program OR plan OR scheme)) AND India) fetched 84 articles. After excluding book chapters, reports and articles with no abstracts or do not refer to India, 62 articles were included and considered for coding. The mapping of the 62 selected articles was undertaken using an MS Excel spreadsheet where articles were placed in the rows and dimensions/taxonomies in the columns. Each article is considered as the unit of coding. A binary coding was undertaken where the elements that are present in the title/abstract/keywords were marked for 1 and absence using 0. An article can be coded into multiple elements under the same dimension in the ontology if it contained those elements. The coding considered the title, abstract and keywords of the articles selected. Coding was performed based on real-time consensus among two of the four authors and an external collaborator who took part in the coding process. 

In the following, we present an assessment of the state of the policies on eliminating NTDs in India and the state of the research on the policies. Subsequently, we present a policy roadmap for the subject and conclude with how the framework and the method can be generalized to other diseases and countries.

## 3. Results

### 3.1. State of the Policies on Eliminating NTDs in India

India experiences the world’s largest absolute burden of at least 11 major NTDs [20]. The Government of India has recognized NTDs as a priority, aiming to eliminate them by 2015, 2018, 2020, and now 2030. However, there has been no systemic approach to systematically eliminating NTDs in India. India’s policies on NTDs may be characterized as follows within the ontological framework:There are a few disease-specific policies/programs that are focused on the elimination of select NTDs primarily to improve the physical health and wellbeing of the target population. They give little attention to other objectives in managing the diseases and outcomes of management in the ontology.The policies/programs see the NTDs dominantly and narrowly as a community health problem to be addressed by personnel from public health institutions.There is a recognition of the very large scale and scope of NTDs in India. Yet, there is little reliable data on them except the selected few that have been targeted for elimination. There is no explicit informational policy to assist eliminating the NTDs.There are many healthcare policies that pay little or no explicit attention to NTDs but could be harnessed to address the challenge.

#### India’s Disease-Specific Policies and Programs

The Government of India is fully dedicated to ending NTDs, such as lymphatic filariasis and kala-azar, in adherence with global elimination and control goals. India has eliminated leprosy in 82% of the cities and districts [43], in addition to infectious trachoma and chronic disease yaws [44]. To further accelerate the elimination of NTDs, the Indian government has implemented several initiatives. These include the Accelerated Plan for Elimination of Lymphatic Filariasis (APELF), a WHO-supported regional alliance established by the governments of India, Bangladesh, and Nepal, the National Rabies Control Programme, the National Vector Borne Disease Control Programme (NVBDCP), the National Leprosy Eradication Programme, and the National Health Policy. The NVBDCP program is for preventing and controlling vector-borne diseases, namely malaria, filaria, kala-azar, Japanese encephalitis (JE), dengue, and Chikungunya. In addition, the National Rabies Control Programme provides vaccination to stray dogs.

India has sought to reduce disease prevalence/incidence in the following ways [24]:Achieve and maintain elimination status of leprosy by 2018, kala-azar by 2023, and lymphatic filariasis in endemic pockets by 2027.To achieve and maintain a cure rate of >85% in new sputum positive patients for TB and reduce the incidence of new cases, to reach elimination status by 2025.Establish regular tracking of the disability-adjusted life years (DALY) index as a measure of burden of disease and its trends by major categories by 2022.

### 3.2. India’s Community Health Approach

To foster India’s community health approach to NTDs, preventive strategies like mass drug administration (MDA) rounds are conducted in endemic areas, wherein anti-filarial medicines are given free of cost to vulnerable communities. Additionally, the government provides support for morbidity management and disability prevention for the population affected by lymphoedema and hydrocele, as well as wage compensation schemes for those suffering from kala-azar and its sequel. India’s Union Health Minister has reiterated the commitment to eliminate lymphatic filariasis by 2027 [45]. 

However, there have been delays in achieving the target due to poor community participation, deep-rooted systemic mistrust issues, lack of champions for the cause, frequent leadership transitions, and postponement of technical advisory committee meetings. In implementation, the delays have been caused by issues such as delays in scheduling mass drug administration rounds, lack of availability of family registers, delays in training plans, and inaccurate counts of community members. 

Mass treatment coverage has also been achieved for people susceptible to filaria. The National Leprosy Eradication Programme was officially declared eliminated as a public health concern in India in 2005 when the new cases fell to less than 1 per 10,000 [46]. Despite this, India still accounts for the largest number of leprosy-affected people in the world.

Consequently, there has been 35% decrease in kala-azar cases reported in 2021 as compared to 2020. In addition, 99% of the kala-azar endemic blocks have achieved their elimination target. This year has taken to the brink of eliminating kala-azar [47]. 

Similarly, MDA coverage and compliance for lymphatic filariasis have improved, with 134 districts having stopped MDA after passing the Transmission Assessment Survey (TAS) [48]. The TAS is a test that determines if the prevalence of lymphatic filariasis in an evaluation area has decreased to a level in which transmission of lymphatic filariasis or recurrence is unlikely even after stopping MDA. 

### 3.3. Scale and Scope of NTDs in India

India has the world’s largest absolute burden of at least 11 major NTDs including hookworm, dengue, lymphatic filariasis, leprosy, visceral leishmaniasis or kala-azar and rabies [20]. Over 670 million people in the country are at risk of infection by Wuchereria Bancrofti and Brugia Malayi parasites in 272 districts, representing around 40% of the global disease burden [47]. Further, India has 23 million people suffering with lifelong disability due to lymphatic filariasis [48]. It is difficult to estimate the actual total burden of all NTDs in India as no single organization or government agency has been given this mandate. The absence of reliable data diminishes the evidence base for the policies, and consequently their effectiveness.

### 3.4. India’s Healthcare Policies and NTDs

The National Digital Health Mission (NDHM) aims to create a digital health ecosystem to support universal health coverage, providing open, interoperable, standards-based digital systems [49]. The mission has 13 objectives, including establishing digital health systems and registries, enforcing open standards, creating personal health records, promoting enterprise-class health applications and portability of health services, using clinical decision support systems, leveraging data analytics and medical research, and strengthening existing health information systems.

Of the 2844 new drugs approved between 1975 and 2022 [50] globally, only 64 (2.25%) were developed explicitly for neglected tropical diseases, even though 11.4% of the global disease burden accounts for NTDs. Only eight new chemical entities were approved for NTDs between 2012 and 2022. Out of 150,000 clinical trials registered between 2011 and 2016, only 1% were for NTDs. These numbers reflect the poor attention NTDs receive. The National Health Policy [24] sets an ambitious goal to meet health needs and ensure that new drugs are affordable and accessible, but it does not discuss in detail NTDs.

The National Policy on Treatment of Rare Diseases emphasises the growing need for research on finding new treatments for rare diseases and infectious diseases [25]. However, the recent draft does not discuss infectious diseases and focuses entirely on rare diseases [25].

The Science, Technology, and Innovation policy [26] does not mention research on NTDs. It aims to bring in the concept of dynamic policy with a robust policy governance mechanism created, communicated, supported, and guided by the Principal Scientific Adviser (PSA), National Institution for Transforming India (NITI) Aayog, and Department of Science and Technology (DST). Further, while the draft National Pharmaceutical Policy, 2017 envisages to create enabling environment to develop and produce innovative drugs [27], the policy does not discuss about drugs related to NTDs. 

The National Biotechnology Development Plan (2021–2025) seeks to encourage one Health Mission on Anti-Microbial Resistance (AMR) for livestock and zoonotic diseases; mission on management and treatment of rare and genetic disorders; establish National Inherited Disorders Administration (NIDAN) Kendras under the Unique Methods Management Inherited Disorders (UMMID) program covering all aspirational districts and promote National Nutrition Mission on Fortified and Functional Foods [51]. It is evident that NTDs has not been the thrust areas of the plan laid out by the Department of Biotechnology, Government of India. NTDs. However, it has reflected upon India’s Coalition for Epidemic Preparedness Initiative (Ind-CEPI) mission to develop low-cost vaccines for endemic preparedness.

The National Intellectual Property Rights Policy (2016) emphasized the need to develop new and affordable drugs by publicly funded R&D institutes and industries but has not laid down a road map pertaining to process and implantation. Regardless, there has been no progress in this area [52].

Nutrition is a vital factor in the process of infection, as it is one part of a complex interplay between four main elements. It sets the context for any strategy to eliminate NTDs, especially given the poor state of nutrition of the population commonly affected by NTDs. The four elements include the virulence of the pathogen, which involves the multiplication and spread of microparasites; the innate susceptibility of the host; and the acquired resistance of the host to the infection, both of which are impacted by nutritional status; and other environmental factors. For the process of infection to be successful, all four of these factors must be considered, with nutrition playing an important role. Nutrition is essential in supplying the host with the necessary sustenance to fight the infection, as well as in providing the nutrients needed for the immune system to function properly [53]. Additionally, nutrition can impact the susceptibility of the host to infection through specific host receptors. For over half a century, India has undertaken several nutrition interventions to reduce malnutrition rates. Examples of these successful steps include the National Food Security Act 2013 [54], National Institute of Public Cooperation and Child Development (NIPCCD), Integrated Child Development Services (ICDS), and the mid-day meal scheme. Nevertheless, despite the strong constitutional and legislative measures of the government, such as policy plans, programme commitments, and supporting institutions, along with a boost in GDP growth, dietary diversity and malnutrition levels remain unchanged. 

### 3.5. Summary of the State of Policies for Eliminating NTDs in India

It is high time for India to establish a comprehensive policy on NTDs that lays down path for greater funding and support for research and innovation on NTDs, and this will be critical for India to achieve the target set in sustainable development goal 3, to end the epidemics of NTDs by 2030 [55]. 

### 3.6. State of the Research on Policies for Eliminating NTDs in India

#### 3.6.1. Monads Map

The monads map in Figure 2 visually and numerically summarizes the frequency of occurrence of each dimension and element of the ontology. The number adjacent to the dimension name and the element is the frequency of its occurrence in the 62 journal articles of India’s healthcare policies for NTDs that were studied and mapped. The bar below the element is a visual indicator of the same, scaled to the maximum number of incidences of any one element. Since a paper may be coded on multiple elements of a dimension, the sum of the frequency of occurrence of elements may exceed the frequency of occurrence of the dimension to which the elements belong. The monads map is described below.

The dominant focus of the very limited research on India’s healthcare policies for NTDs has been on the objective (57), disease (57), and policy (52). There is substantially less focus on the outcome (33) and there is less focus on the entity (31).

A substantial proportion of articles consider different objectives for India’s healthcare policies for NTDs. The dominant focus is on elimination (25) and control (25), followed by prevention (20). There is medium focus on cure (18), identification (16), and education (14). The least focus is on eradication (5), palliation (2), and rehabilitation (1). Specific objectives were given some focus in the research, whereas there was relatively little on emerging healthcare needs for rehabilitative care and palliative care.

Although all the 62 articles are related to NTDs, only 57 specify the type of disease. The dominant focus is on infectious helminth (21) diseases. The next significant emphasis is on infectious bacterial (17) and infectious protozoal (16) diseases. Diseases such as co-infectious (6), infectious viral (5), and non-infectious snake bites (3) are given some attention. There is very little focus on infectious fungal (1) and infectious ectoparasitic (1), and no mention of non-infectious sickle cell anaemia.

The research covers a spectrum of policy mechanisms and is heavily focused on executive (28), informational (16), and R&D (15) policy mechanisms. There is little emphasis on legislative (8), fiscal/financial (6), economic (3), social (3), and legal (1) policy mechanisms. There is no mention of policy mechanisms that are constitutional (0), regulatory (0), and contractual (0).

The research focuses on different outcomes, but it is dominantly focused on the economic (15) wellbeing followed by physical health (10). There is equal focus on outcomes such as social (9) and societal (9) wellbeing. The least emphasized outcomes are mental health (3) and environmental (2) wellbeing.

The research focuses least on the entities that are or potential agents of elimination of NTDs in India. Among the different entities, it largely focuses on the public institution (15). There is less focus on private institution (8), personnel-doctor (7), community (7), academia (6), and institution-NGOs (6). The other entities such as personnel-paramedics (3), personnel-nurse (3), personnel-staff (3), family (3), and individual (2) have not been given much attention in the research.

#### 3.6.2. Themes Map

The themes map visually summarizes the co-occurrence of elements of the ontology in the population of articles, as shown in Figure 3. 

The primary research theme (in red) is the executive mechanisms for control of infectious helminth diseases. It is a short segment of many potential pathways for elimination of NTDs in the ontology, which include executive mechanisms for control of infectious helminth diseases. The potential pathways may include the participation of any one or more entities for achieving any one or all the outcomes for wellbeing. Further, among the policy dimensions, it only includes one, and among the objective and diseases dimensions, it includes one, respectively (i.e., control and infectious helminth). The primary theme is one dimensional and one-levelled—it is simple.

The secondary research theme (in brown) is the objective for the identification, prevention, cure, and elimination of infectious bacterial diseases. It represents four short segments of many potential pathways for the elimination of NTDs in the ontology. They are the identification of infectious bacterial diseases, prevention of infectious bacterial diseases, cure of infectious bacterial diseases, and elimination of infectious bacterial diseases. Further, these potential pathways may include different policy mechanisms, and the involvement of one or many entities for achieving one or all of the outcomes.

The tertiary research theme (in yellow) is informational, research, and developmental (R&D) mechanisms for education on infectious protozoal diseases. It represents two short segments of many potential pathways in the ontology-informational mechanisms for education on infectious protozoal diseases and R&D mechanisms for education on infectious protozoal diseases. These potential pathways may include one or more entities for achieving one or more outcomes.

The quaternary research theme (in blue) is the role of policy mechanisms (like legislative and financial mechanisms) relating to infectious viral and co-infectious diseases controlled by entities (like public institutions, private institutions, NGO institutions, and community) for outcomes in terms of physical health and economic, social, and societal wellbeing. 

The quinary research theme (no colour) summarizes the absence in the research corpus. Constitutional, economic, and regulatory mechanisms are not part of any theme, although legislative and executive mechanisms which are part of the quaternary and primary themes, respectively. The research on the role of contractual, legal, and social mechanisms for devising healthcare policies for NTDs in India are absent in the corpus. Among the objective, thematic focus on palliation, rehabilitation, and eradication was missing, although there is inclusion of control, identification, prevention, cure, and elimination of NTDs. Similarly, thematic inclusion of infectious fungal and ectoparasitic diseases, non-infectious snake bites and sickle cell anaemia were absent.

Amongst the entities, there is no thematic inclusion and systematic distinction between the role of academia, personnel (doctors, paramedics, nurses, and staff), family and individuals in the research. It also highlights the absence of systematic consideration of mental health wellbeing and environmental wellbeing in the research. The quinary theme is six-dimensional and many-levelled, and there is a vast, complex, unresearched domain.

### 3.7. Summary of State of the Research on Policies for Eliminating NTDs in India

The research on policies for eliminating NTDs in India is very limited. The quantum of research is miniscule in relation to the scale and complexity of the challenge. It is inadequate to systematically (a) describe effective systemic policy pathways to address the challenge, (b) explain the rationale behind the policies, (c) predict the outcomes of implementing the policies, and (d) control the outcomes through feedback and learning. The siloed, selective, and segmented research makes it difficult to design effective pathways, redirect ineffective pathways, and research novel ones.

## 4. Discussion—Policy Roadmap for Eliminating NTDs in India

The drive to eliminate NTDs in India must be a timebound, concerted, comprehensive, adaptive, national effort that breaks the past pattern of select targets and frequently postponed target dates. There must be a roadmap navigated with continuous feedback and learning to guide the actions along effective pathways, away from ineffective pathways, and for exploration of innovative pathways.

The Government of India (GoI) must establish a purposive, proactive system for the drive. The system must be governed by a standing national committee and state/union territory committees to formulate a systemic agenda for systematic research, policies, and practices for eliminating NTDs. These committees must be responsible and accountable for the outcomes. The national committee’s agenda must inform and be informed by the constituent state and union territory committees’ agendas, and those of the world bodies such as WHO, UN agencies, and others.

The ontology of healthcare policies to eliminate NTDs (or a similar framework) must be adopted as a systemic national framework for all the states and union territories. The framework will be critical to integrate the effort nationally, address the challenge, and provide a roadmap. Within the framework, each state must choose its pathways based on its local requirements, priorities, knowledge, and resources. Given the global disease burden and the vision of achieving Sustainable Development Goal 3.3, India could adapt strategies to enhance drug and vaccine development [15], improve its diagnostics and deploy new technologies [19], and prioritize health research [56] and initiatives for control and elimination of neglected tropical diseases. The adoption of a common framework will also help formalize and transfer the knowledge about, feedback from, and learning from the implementation from across the country, between the states and union territories, and from other countries. This systematic approach will help move the cycle of generation and application of knowledge on the challenge from a selective, segmented, and siloed effort to a synoptic, systemic, and systematic one.

The state of the art, state of the need, and state of the practice of elimination of NTDs by states and union territories must be periodically mapped on to the framework. Analysing the gaps between the three states of knowledge must guide the translation of research to policy to practice and then back to research, for virtuous feedback and learning cycles to achieve the vision.

Thus, the national GoI Committee must help the states and union territories collaborate, coordinate their policies, and communicate their learning. It must set the trajectory for elimination of NTDs in the states and nationally. It must provide a ‘Google Map’ for the nationwide effort, with timely correction, redirection, and innovation.

The following core policy recommendations are organized by the columns of the ontology and then integrated. These recommendations are aligned with the present research, policies, practices, and recommendations in India and of the WHO, UN agencies, and other similar bodies.

### 4.1. Policies for Mapping the NTDs in India

The lack of a comprehensive medical registry on NTDs presents significant challenges for healthcare providers and public health officials. Without an accurate understanding of the burden and prevalence of these diseases, it is impossible to properly allocate resources for NTD prevention and control. Moreover, the lack of a registry hinders the ability to track the effectiveness of existing interventions and identify risk factors for NTDs. Additionally, it makes it difficult to assess the impact of NTDs on individuals, communities, and health systems. Furthermore, the lack of an accurate registry limits the development of new diagnostics and treatments as well as the ability to determine the best strategies for controlling and eliminating NTDs. It is therefore essential that a medical registry be established to ensure that all stakeholders have access to the most up-to-date and accurate information on NTDs. Such a registry would enable healthcare providers, public health officials, and researchers to better address the burden and prevalence of these diseases and develop more effective strategies for their prevention and control. The following research highlights the necessity of mapping the NTDs in India.

There is growing interest and commitment to the control of schistosomiasis and other so-called NTDs. Resources for control are inevitably limited, necessitating assessment methods that can rapidly and accurately identify and map high-risk communities so that interventions can be targeted in a spatially explicit and cost-effective manner. Research efforts should be undertaken to determine the optimal strategy of rapidly and simultaneously assessing several NTDs. An immediate question that arises is whether it is possible to develop an integrated rapid mapping approach [57].A sound understanding of NTD distribution and prevalence is an essential prerequisite for cost-effective control, with each national programme needing to be tailored to its specific context. Appropriate targeting of integrated MDA requires information on the geographical distribution of different NTDs to identify areas that would benefit most from this approach. In the absence of such information, priority areas for control or elimination and the estimation of drug requirements is often based on expert opinion or out-of-date information [58].There is an urgent need for better surveillance and disease burden assessments for most of the NTDs, but especially for amebiasis, leptospirosis, and the major arbovirus infections, and for linking mass drug administration, vaccinations, integrated vector management, and improved surveillance together as part of overall efforts to strengthen health systems in the South Asian region [59].

The current map of NTDs in India is incomplete and inadequate. Inadequate epidemiological data and failure in translation of basic research into new technologies remains a significant gap in India’s approach towards NTDs [16]. Detailed and reliable maps are needed for navigation nationally and locally. The NTDs must be mapped geographically, demographically, temporally (seasonally and historically), environmentally, economically, and socially. These maps must be updated periodically to be current. These data on the NTDs must be the foundation of precise policies to eliminate NTDs in India.

An ongoing census of the NTDs should form the baseline for the policies to eliminate NTDs in India. The census should be the basis for determining the scale and scope of the effort to eliminate an NTD and the synergy between the efforts to eliminate them. For instance, the application of a mobile-based management information system in rural and urban parts of Tanzania aided in control, accurate classification, diagnosis, and assessment of childhood illness [60].The census should describe, in depth and in detail, the prevalence of NTDs in India. It should be the empirical basis for explaining the causes and consequences of the NTDs individually and in aggregate. It should be used to predict the trajectory of the NTDs and their effects. Last, it should be used to control their trajectory through continuous feedback and learning.The digital integration and use of technology are critical for devising sustainable surveillance response systems for NTDs [61]. For instance, Brazil created LeishCare (a mobile application which recorded patient information) to facilitate diagnosis, management, and timely treatment of patients with NTDs [62]. In addition, systems like Brazil’s electronic health records and technology ecosystem for COVID-19 response would facilitate the integration [63,64].India’s emerging digital infrastructure for healthcare must be used as the platform to manage the census of the NTDs in India.The census in conjunction with the medical, local, and personal knowledge of the entities in the ontology should determine the objectives regarding each NTD and the desired outcomes of fulfilling the objectives.The census should be the basis for local differentiation and national integration of the policies to eliminate NTDs.The census should be the basis of differentiation by diseases and integration across class of diseases. Digital public health paradigm in Africa, for instance, highlighted the importance of mHealth, eHealth, and electronic health records for improving health service access, service delivery, and monitoring outcomes in surveillance of infectious diseases [65].The above factors are critical for leveraging India’s National Digital Health Mission (NDHM) towards creating citizen centric holistic healthcare program [49]. In addition, India’s Health Data Management Policy, 2020 could be a potential catalyst for creation of digital health systems.

### 4.2. Policy Objectives for Managing the NTDs in India

The ultimate objectives regarding NTDs are to eradicate them where they prevail presently, eliminate them so that they do not recur, and control the conditions to prevent their reemergence. The history of management of diseases, both neglected (for example, leprosy) and not neglected (for example, smallpox), shows the difficulty of achieving these objectives. The earlier objectives in the ontology—identification, prevention, cure, palliation, rehabilitation, and education—can significantly contribute to the outcomes in a shorter timeframe and at a lower cost. These objectives must be prioritized by diseases based on the state of the knowledge, policies, and practices on managing each disease. The following research highlights the necessity of managing the NTDs in India.
Snakebite envenoming requires establishment of dedicated rehabilitation programs, addressing both psychological and physical disability, will improve the recovery of survivors, enabling more of them to return to useful, productive lives, therefore increasing economic productivity [66].Current NTDs control is not fully comprehensive, and it contributes towards the preventative and curative strands of Universal Health Coverage, but the rehabilitative and palliative aspects of NTDs, and the intersection between NTDs and disability, are not well prioritized [67].First-line health services are for many people the first entry point into the health care system. The integration of the various care and service providers is important from an efficiency point of view. The provision of integrated care, i.e., provision of curative and preventive care at first line, is desirable for its effect on effectiveness [68].

In this context, the objectives for managing the NTDs in India must be clearly defined. The identification of many NTDs is difficult but must be the highest priority for all NTDs. It is essential for an accurate census. It must be a part of the agenda of all the entities in the ontology, with suitable educational support for them, and infrastructure (digital and non-digital) for them to record the same. Despite the high rate of prevalence of sickle cell anaemia among the tribal population of India [69], it has been neglected in the themes map (Figure 3). Similarly, snake bites that are highly prevalent in India are not emphasized in the themes map. There is a need for systematic priority setting and targeted policies to ease the disease burden [70].The prevention of NTDs will be disease-dependent and may depend on one or a combination of entities. Its priority may be based on the scale and scope of prevalence, ease of prevention, and the impact on wellbeing. Preventive chemotherapy has been substantially used for treatment and control of NTDs. However, the need for improved diagnostics for NTDs is critical for devising treatment strategies across levels of control, interruption of transmission, elimination, and post-elimination surveillance [19].Prevention policies may be medical (for example, vaccination), personal (for example, habits), societal (for example, practices), or environmental (for example, water contamination). For instance, a dracunculiasis eradication program mainly focuses on the preventive measures such as access to improved drinking-water sources, improved surveillance, encouraging self-reporting, and preventing infected individuals from swimming in drinking water sources, active case surveillance and vector control [71].Curing NTDs will depend on the availability, accessibility, quality, and cost of healthcare available for a disease. Many NTDs do not have a cure. Policies on cure must address the design, development, and delivery of drugs to cure a disease. The priority of a cure for an NTD should be based on the above factors.The palliation of the effects of an NTD may be an alternative if a cure is unavailable, or to manage the aftereffects of a cure. Its priority for an NTD will be determined by the efficacy of cure (if available), aftereffects of the cure, or the effects of the disease itself irrespective of the cure.The rehabilitation of a person that has suffered from an NTD, with or without it being cured, would depend on the residual effects of the disease and the treatment. The policy on rehabilitation must address the effects on the economic, social, and societal wellbeing of the individual, his/her family, and the community. The themes map (Figure 3) also reflects the least emphasis placed on palliation and rehabilitation, which could be due to the absence of institutional mechanisms and follow-up actions [22].The education of entities is central to all the other objectives, and all the outcomes. Policies on education must be comprehensive and contextual. They must be differentiated by the combinations of objective–disease–entity–outcome but integrated across all of them. The identification of and education on infectious NTDs [72,73] are critical.Eradication is the publicly stated objective for NTDs, but it is difficult to achieve. The policies regarding the earlier objectives are necessary but not sufficient for eradication. The cost of identifying and eradicating the last pocket of a neglected disease can be prohibitive. The policies must address identification based on the census of the diseases and the subsequent steps necessary for elimination. Devising policies with strategies for the control of single diseases and multiple infectious diseases by involving medical personnel [74], community participation, government–community partnerships, private–public partnerships (PPPs), not-for-profit sector, civil societies, industries, and health services research [38] are critical for the elimination of NTDs.Elimination policies may pose an even greater challenge than eradication. They will entail locating and eliminating the last genetic stores of the diseases.Control policies may be aimed at managing the occurrence of an NTD, its spread, and its recurrence. They may be aimed at the biological agents of a disease, the environmental enablers of a disease, or the entities responsible for managing the disease.

### 4.3. Policy Outcomes of Managing the NTDs in India

The six outcomes are both independent of and interdependent on each other. Consequently, there is always a trade-off between them that needs to be managed. For example, maximizing physical health wellbeing may reduce economic wellbeing. Harmonizing the different outcomes with varying time horizons will depend on the choice of objectives for the prevalent diseases and the priorities of the entities affected by it.

People affected by NTDs are frequently the target of social stigmatization. Leprosy is an obvious example and has been a major focus of stigma studies in the past. The psycho-social aspects of stigmatization associated with disfiguring NTDs is well documented, but stigma is also an important social determinant of the effectiveness of disease control because of its impact on help-seeking and treatment adherence. Furthermore, it has been shown that stigma influences political commitment to NTD control [75].

The priority of outcomes for an NTD or a class of NTDs must be differentiated by NTD geography and integrated across the geographies. ○The health (physical, mental), economic, environmental, social, and societal priorities for an NTD or class of NTDs must be geography-specific. Contextual (intrinsic and extrinsic) determinants of NTDs [37] must be considered for comprehensive outcomes.
The priority of outcomes for an NTD or a class of NTDs must be aligned with the evidence about the disease, experience with managing the disease, and the objective of managing it.
○The health (physical, mental), economic, environmental, social, and societal priorities for an NTD or class of NTDs must be disease- or disease-class-specific.
The priority of outcomes for an NTD or a class of NTDs must be aligned with the priorities of the entities affected by it. ○The health (physical, mental), economic, environmental, social, and societal priorities for an NTD or class of NTDs must be specific to the corresponding individual, family, and community. For instance, educational, psychosocial, medical, and residential support were found to be major drivers for formulating social sustainability measures for people affected with leprosy [76].


### 4.4. Policies on Entities for Managing the NTDs in India

The elimination of NTDs must be the responsibility of the full spectrum of entities based on their role, requirements, and the need they satisfy. It must be a concerted, coordinated effort from the entities. 

Evidence shows integration works best when aimed at people with severe, complex, and long-term needs. It offers a new opportunity for managing morbidity and long-term disabilities in the community, through greater coordination between health, social and community care. This is something that has not yet gained wide attention from the NTD community [77]. 

Innovative partnerships, such as Uniting to Combat NTDs, which has produced critical information on NTDs and NTD-related advocacy, will be essential for moving forward the response. Establishing and strengthening links between NTD advocacy groups and social justice organizations can leverage and expand these partnerships. Increased linkages also promote cross-sector accountability within the partners, between sectors, and across human rights mechanisms [78].

The integration of surveillance and interventions will not be possible without considerable political support, at several levels. There will be challenges towards integration, including relationships with donors, potential changes to NTD management structures, and complexities in health care worker training among many others, and any of these challenges could derail efforts to achieving integrated management. Strong relationships will be required between governments, international agencies, implementing partners, and donors, with a clear plan of action supported by an evidence base to move forward an agenda of integration [79].

All the entities must be empowered and enjoined to eliminate NTDs in India through their roles and responsibilities. The agencies of the entities must be differentiated and integrated accordingly.They should be proactive stakeholders and participants in conducting the census of the NTDs, formulating objectives for managing them, specifying the desired outcomes, and formulating the policies.They should be proactive generators and practitioners of both explicit and tacit knowledge about the NTDs, corresponding to their roles and responsibilities. The role of academia and public laboratories for engaging in drug innovation and repositioning is critical for India [80].

### 4.5. Policies for Managing the NTDs in India

A wide range of policies must be deployed to assure the elimination of NTDs in India. The policies may be legislative, economic, regulatory, fiscal/financial, informational, contractual, legal, and social. The choice of policies must be comprehensive but contextual: highly differentiated by diseases, objectives, and outcomes, but tightly integrated across them.

The specification of policies on elimination of NTDs must be based on a synthesis of the state of the research, policies, and practices particular to the requirements and the context. This rich body of knowledge must be synthesized through the lens of the ontology to determine the pathways that have been effective, ineffective, and unexplored.

Human rights advocacy on NTDs can bring attention to the devastating effects of NTDs and the ongoing resource gap for disease prevention and treatment. From a programmatic perspective, rights-based advocacy and engagement with human rights mechanisms could support more comprehensive structural responses that can lead to sustainable NTD control and elimination results [78].

Each type of policy can play a role in achieving the objectives for a disease and obtaining the desired outcomes through the agency of the entities. The roles of the policies must be understood, differentiated, and integrated. For example, an executive policy for the prevention of a disease by medical personnel may have to be deployed in conjunction with social/informational policies for the education of a community to be effective.

A policy may provide an impetus to achieve an objective, set the norm for doing so, or even be a barrier to its achievement. For example, an executive policy for the eradication of a disease may (a) drive academic research, medical personnel practice, and institutional actions, (b) set the norms/expectations for different time horizons, and (c) impede other objectives such as palliation and rehabilitation. The legislative policy must be implemented in conjunction with social policies aimed at the community/family/individual for social and societal wellbeing.

Now, some policies have discussed the importance of developing new and affordable drugs through publicly funded R&D institutes and industries. Despite this, no concrete roadmap has been developed to facilitate this goal. This has led to a lack of progress in this area. To address this, a clear plan needs to be formulated to outline the steps necessary for achieving this goal. This could include measures such as incentivizing research and development, providing necessary resources, and establishing regulations to protect intellectual property rights. Additionally, public–private partnerships should be explored to facilitate the collaboration between public and private institutions. Furthermore, research and development initiatives should be regularly monitored to ensure that they are making progress. By implementing such measures, it will be possible to increase access to new and affordable drugs.

## 5. Conclusions

The paper assesses the present state of the policy and policy research on NTDs in India through the lens of the ontology of healthcare policies to eliminate NTDs in India. The ontology, like a good theory, (a) describes the elements of the problem, (b) explains the potential pathways through the combination of elements, (c) can be used to explain the effectiveness of the pathways, and (d) controls the trajectory of the policies through feedback about the effectiveness of the pathways and learning from the feedback. 

The paper systematically identifies the gaps in NTD policies, and research on those policies. It proposes policy guidelines derived from the ontology to systematically address the challenge. Thus, the paper provides a roadmap to address a vexing, longstanding problem of India. It can be used like a ‘Google Map’ to navigate the complexity of navigating the elimination of NTDs. 

The framework and the method of analysis can be generalized to provide roadmaps for other countries facing a similar challenge, and for other diseases of similar complexity. The roadmaps, with continuous feedback and learning, can help navigate the challenge efficiently and effectively.

## Figures and Tables

**Figure 1 ijerph-20-06842-f001:**
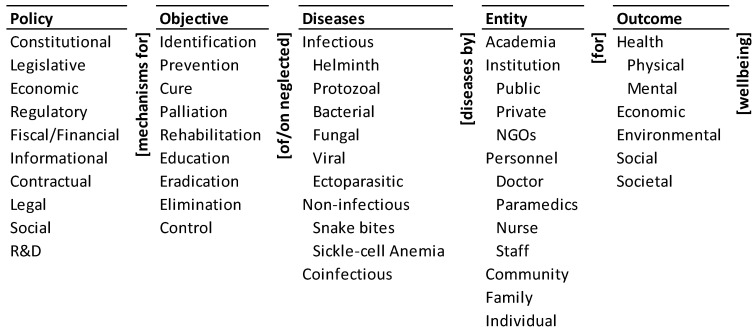
Ontology of healthcare policies to eliminate NTDs in India.

**Figure 2 ijerph-20-06842-f002:**
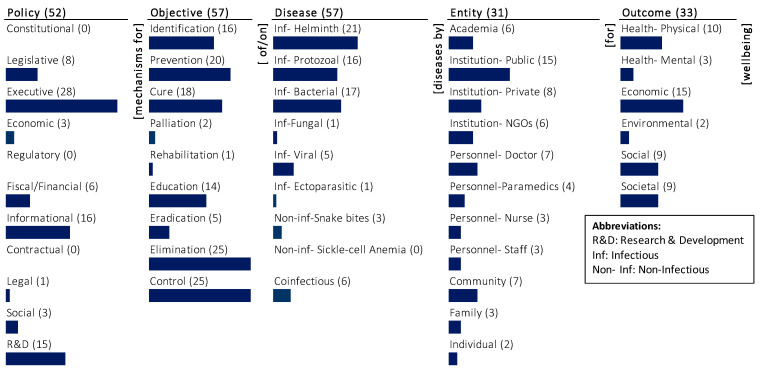
The monads map of research on policies for eliminating NTDs in India.

**Figure 3 ijerph-20-06842-f003:**
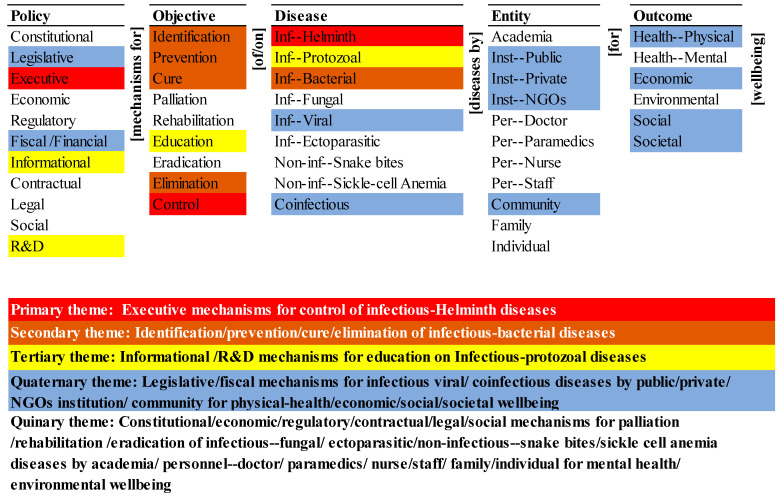
Themes in the research on policies for eliminating NTDs in India.

## Data Availability

Not applicable.

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
