# Peer review of "Healthcare Policies to Eliminate Neglected Tropical Diseases (NTDs) in India: A Roadmap"

_ijerph, 2023, doi:10.3390/ijerph20196842_

Round 1

Reviewer 1 Report

l   The authors have not clearly articulated the conceptual framework for the Ontological framework. Is it referenced from a specific literature source, or is it developed independently? If it is the former, proper citation of the literature should be provided. If it is the latter, a detailed explanation of the development process should be described.

l   The sources of research results such as 'State-of-the-Policies and State-of-the-Research on Policies for Eliminating NTDs in India' have not been described in the materials and methods section. The latter is discussed in the results section, which does not adhere to the standard principles of academic journal writing.

l   The discussion content tends to be overly generalized. There is a need to strengthen the discussion, particularly regarding the gap between policy and research highlighted in the results.

l   The authors have used 'roadmap' as the central concept of this paper. However, whether in the results or the discussion, the description of the roadmap was not clear. It is recommended that the authors provide a clearer explanation of the roadmap. If the authors do not intend to make this modification, it is suggested that they remove 'roadmap' from both the title and the content of the manuscript.

Author Response

a) The authors have not clearly articulated the conceptual framework for the Ontological framework. Is it referenced from a specific literature source, or is it developed independently? If it is the former, proper citation of the literature should be provided. If it is the latter, a detailed explanation of the development process should be described.

Thank you for catching our oversight. The ontology is based on our past work and other literature. We have cited the corresponding literature.

b) The sources of research results such as 'State-of-the-Policies and State-of-the-Research on Policies for Eliminating NTDs in India' have not been described in the materials and methods section. The latter is discussed in the results section, which does not adhere to the standard principles of academic journal writing.

Thank you for catching our omission and misplacement. We have added a section on the policies and practices to the materials and methods section. We have also moved the corresponding description for the research on policies from the results section.

c) The discussion content tends to be overly generalized. There is a need to strengthen the discussion, particularly regarding the gap between policy and research highlighted in the results.

We agree that it is long – but a complex, long-standing, important problem probably requires an extended discussion. We have introduced several issues that have hitherto not been considered or given adequate attention. We have erred on the side of more explanation rather than less. As we have mentioned in the manuscript: “The following core policy recommendations are organized by the columns of the ontology and then integrated. These recommendations are aligned with the present research, policies, practices, and recommendations in India and of WHO, UN agencies, and other similar bodies.” On many of these issues the literature is thin. We have tried to be as specific as possible given the literature and without overextending it. We hope you agree.

d) The authors have used 'roadmap' as the central concept of this paper. However, whether in the results or the discussion, the description of the roadmap was not clear. It is recommended that the authors provide a clearer explanation of the roadmap. If the authors do not intend to make this modification, it is suggested that they remove 'roadmap' from both the title and the content of the manuscript.

We appreciate your comment but would like to retain the characterization as a ‘roadmap’ for the following reasons. Perhaps the semantic confusion is between a procedural roadmap and a substantive one. What we present is a procedural roadmap for how to address the challenge systemically and systematically using the ontology as a framework. The substantive roadmap, the actual pathways to be reinforced, redirected, and researched must flow out of this effort. It would be difficult and premature to propose a substantive roadmap at this stage, given the paucity of research, policies, and practices. I hope you agree.

Reviewer 2 Report

The manuscript "Healthcare Policies to Eliminate Neglected Tropical Diseases (NTDs) in India: A Roadmap" proposes to present an ontological framework for public policy aimed at contributing to the response to neglected tropical diseases in India.

Neglected tropical diseases are a global public health problem, despite many efforts there is still no effectiveness in adequately responding to this problem. The authors rightly highlight this problem and relate it to socioeconomic issues. Countries with greater inequality and poverty are more affected by neglected tropical diseases.

The manuscript presents an important theme of global interest, despite being contextualized in India. Countries like Brazil, for example, still have a lot of problems in relation to neglected tropical diseases, such as leprosy, dengue, zika virus, leishmaniasis, schistosomiasis and others. Therefore, this manuscript is a contribution, as it makes room for important discussions, which cannot be forgotten by health authorities in the global context. It is certainly necessary to do more and innovate in this field, because the results are not yet satisfactory.

Recommendations

Introduction

[1] The introduction of the manuscript is well written and very easy to read, however the authors do not make the objectives of the manuscript very clear. It is necessary to clarify the objectives, for example, better explain the concept of ontology, in the introduction this is not exactly described in depth, but it is clear what it means.

[2] It is important for the authors to discuss a little more in the Introduction about neglected diseases in the BRICS countries (Brazil, Russia, India, China and South Africa). I consider it important to bring data on neglected tropical diseases in these countries. Likewise, this type of information does not need depth in the Introduction, however their contextualization is important, as it helps to reinforce the authors' arguments.

Materials and Methods – Ontology of Healthcare Policies to Eliminate Neglected 114 Tropical Diseases (NTDs) in India

[3] The authors make this statement "Achieving the objectives would require different constitutional, legislative, eco- 124 economic, regulatory, fiscal/financial, informational, contractual, legal, social, and Research 125 and Development (R&D) policy mechanisms". So I ask the following question: are cultural issues not part of this agenda? Authors should consider this question and if they deem it necessary to reformulate this sentence.

[4] Figure 1 is very good, but the authors should explain it better in the text. Make it clear that the reading is by column. Is there a relationship between the columns? How are they related?

[5] "The ontology expresses 10*9*9*11*6 = 53,640 such pathways with potentially multiple 152 instantiations of each." The issue of combinatorial analysis is very clear, but when it comes to public health policies, this is not entirely true. Many of the ways to be pointed out should consider budget, infrastructure issues, technical capacity and socio-cultural issues. I recommend that authors use references to support this argument and also improve writing, because the way it is the text is confused without scientific basis.

[6] The authors need to better describe the methodology, as it is not possible to understand the results from what was written. I read the methodology and found something else in the results.

"State-of-the-Research on Policies for Eliminating NTDs in India"

[7] This excerpt should be in the methodology of the article.

"We searched for the literature using the research indexing service Elsevier’s Scopus.
 The search term TITLE-ABS-KEY ((neglected W/5 disease AND (policy OR program OR plan OR scheme)) AND India) fetched 84 articles. After excluding book chapters, reports and articles with no abstracts or do not refer to India, 62 articles were included and considered for coding. Mapping of the 62 selected articles was undertaken using an MS Excel spreadsheet where articles were placed in the rows and dimensions/taxonomies in the columns. Each article is considered as the unit of coding. A binary coding was undertaken where the elements that are present in the title/abstract/keywords were marked for 1 and absence using 0. An article can be coded into multiple elements under the same dimension in the ontology if it contained those elements. The coding considered the title, abstract and keywords of the articles selected." 

Observation

[8] Epidemiological scenario in India regarding neglected tropical diseases. When starting to read the discussion section, I realized that the authors could create a subsection, or include in the introduction of the article an epidemiological scenario in relation to neglected tropical diseases in India. This point is portrayed throughout the article, but without presenting the epidemiological scenarios, with the prevalence of these diseases in the population.

Discussions

[9] The authors make the following claim: "No similar guiding framework currently exists in any country; India's will be the first". However, the authors cannot support this statement, as their studies do not provide them with this data. In this case, I recommend that authors rewrite that sentence, or include a related work section that highlights what other countries in the world are doing.

[10] The authors cite in the recommendations examples of health information systems and digital health technologies, which can help improve the response to neglected tropical diseases. As the authors cite an example from Brazil, I recommend reading the following manuscript: "Electronic health records in Brazil: Prospects and technological challenges". I believe that the authors will be able to perceive that there are similarities between Brazil and India in this process of digital transformation of health. Another manuscript that I also leave as a recommendation is "The relevance to a technology ecosystem in the Brazilian National Health Service’s covid-19 response: the case of Rio Grande do Norte, Brazil".

[11] Discussion session is good as it includes several aspects, but it is very long and sometimes confusing. I believe that it would be interesting for the authors to better structure the discussions, I believe that this section can be based on the ontology map, but this is only a suggestion.

[12] I did not notice anything related to public funding in the discussions. Another issue, the authors talk about education and information, which is pertinent, however, the authors could go deeper into this aspect. In Brazil, the AVASUS Platform (Virtual Learning Environment Sistema Único de Saúde do Brasil) has been used in several public health crises, for example, in response to the syphilis epidemic and in the prison system.

Minor editing of English language required

Author Response

[1] The introduction of the manuscript is well written and very easy to read, however the authors do not make the objectives of the manuscript very clear. It is necessary to clarify the objectives, for example, better explain the concept of ontology, in the introduction this is not exactly described in depth, but it is clear what it means.

Thank you for your compliment. We have modified the introduction to clarify the objectives. We hope it is satisfactory.

[2] It is important for the authors to discuss a little more in the Introduction about neglected diseases in the BRICS countries (Brazil, Russia, India, China and South Africa). I consider it important to bring data on neglected tropical diseases in these countries. Likewise, this type of information does not need depth in the Introduction, however their contextualization is important, as it helps to reinforce the authors' arguments.

Thank you for the suggestion. We have included a short statement with citations.

[3] The authors make this statement "Achieving the objectives would require different constitutional, legislative, eco- 124 economic, regulatory, fiscal/financial, informational, contractual, legal, social, and Research 125 and Development (R&D) policy mechanisms". So I ask the following question: are cultural issues not part of this agenda? Authors should consider this question and if they deem it necessary to reformulate this sentence.

Thank you for your observation. Cultural issues are part of the agenda. As explained in response to [5] below they can be drivers of, norms for, and barriers to the pathways. We have considered them separately in the papers cited. However, here, we would like to consider it as exogenous to the feasibility and effectiveness of the pathways rather than as being endogenous to them.

[4] Figure 1 is very good, but the authors should explain it better in the text. Make it clear that the reading is by column. Is there a relationship between the columns? How are they related?

Thank you. We have added the following sentence: “Each pathway is a concatenation of an element from each column of the ontology, with the adjacent connecting words/phrases.” We hope that clarifies the relationship between the columns.

[5] "The ontology expresses 10*9*9*11*6 = 53,640 such pathways with potentially multiple 152 instantiations of each." The issue of combinatorial analysis is very clear, but when it comes to public health policies, this is not entirely true. Many of the ways to be pointed out should consider budget, infrastructure issues, technical capacity and socio-cultural issues. I recommend that authors use references to support this argument and also improve writing, because the way it is the text is confused without scientific basis.

Thank you for your comment. We have tried to clarify the construction of the logical pathways and their practical limitations as suggested by the reviewer. We agree that budgetary, infrastructural, technical, socio-cultural, and other similar issues can be drivers of, norms for, or barriers to the effectiveness of the pathways. We have also cited our papers that explicitly consider the issues mentioned. However, here, we would like to consider them as exogenous to the feasibility and effectiveness of the pathways rather than as being endogenous to them.

[6] The authors need to better describe the methodology, as it is not possible to understand the results from what was written. I read the methodology and found something else in the results.

We have added two sub-sections to the materials and methods section. I hope the changes clarify the basis of the results.

[7] This excerpt should be in the methodology of the article.

"We searched for the literature using the research indexing service Elsevier’s Scopus.  The search term TITLE-ABS-KEY ((neglected W/5 disease AND (policy OR program OR plan OR scheme)) AND India) fetched 84 articles. After excluding book chapters, reports and articles with no abstracts or do not refer to India, 62 articles were included and considered for coding. Mapping of the 62 selected articles was undertaken using an MS Excel spreadsheet where articles were placed in the rows and dimensions/taxonomies in the columns. Each article is considered as the unit of coding. A binary coding was undertaken where the elements that are present in the title/abstract/keywords were marked for 1 and absence using 0. An article can be coded into multiple elements under the same dimension in the ontology if it contained those elements. The coding considered the title, abstract and keywords of the articles selected."

You are right. The above was misplaced. It has been moved to the materials and methods section.

[8] Epidemiological scenario in India regarding neglected tropical diseases. When starting to read the discussion section, I realized that the authors could create a subsection, or include in the introduction of the article an epidemiological scenario in relation to neglected tropical diseases in India. This point is portrayed throughout the article, but without presenting the epidemiological scenarios, with the prevalence of these diseases in the population.

We would appreciate your indulgence in keeping the epidemiological discussion unchanged. The policies and the corresponding ontology are the focus of the paper. Hence we have woven the epidemiological discussion in the background of the discussion of the policies and ontology, at suitable points.

[9] The authors make the following claim: "No similar guiding framework currently exists in any country; India's will be the first". However, the authors cannot support this statement, as their studies do not provide them with this data. In this case, I recommend that authors rewrite that sentence, or include a related work section that highlights what other countries in the world are doing.

Thank you for the observation. We have deleted the sentence.

[10] The authors cite in the recommendations examples of health information systems and digital health technologies, which can help improve the response to neglected tropical diseases. As the authors cite an example from Brazil, I recommend reading the following manuscript: "Electronic health records in Brazil: Prospects and technological challenges". I believe that the authors will be able to perceive that there are similarities between Brazil and India in this process of digital transformation of health. Another manuscript that I also leave as a recommendation is "The relevance to a technology ecosystem in the Brazilian National Health Service’s covid-19 response: the case of Rio Grande do Norte, Brazil".

Thank you for the suggestions. We have added a statement and cited the two papers.

[11] Discussion session is good as it includes several aspects, but it is very long and sometimes confusing. I believe that it would be interesting for the authors to better structure the discussions, I believe that this section can be based on the ontology map, but this is only a suggestion.

We are glad you liked the discussion. We agree that it is long – but a complex, long-standing, important problem probably requires an extended discussion. We have introduced several issues that have hitherto not been considered or given adequate attention. We have erred on the side of more explanation rather than less. As we have mentioned in the manuscript: “The following core policy recommendations are organized by the columns of the ontology and then integrated. These recommendations are aligned with the present research, policies, practices, and recommendations in India and of WHO, UN agencies, and other similar bodies.” In fact, the discussion is based on the ontology as you have suggested and integrated with it. We hope you agree.

[12] I did not notice anything related to public funding in the discussions. Another issue, the authors talk about education and information, which is pertinent, however, the authors could go deeper into this aspect. In Brazil, the AVASUS Platform (Virtual Learning Environment Sistema Único de Saúde do Brasil) has been used in several public health crises, for example, in response to the syphilis epidemic and in the prison system.

Thank you for the observation. We hope the explanation with reference to [5] above addresses the issue of inclusion of public funding explicitly.

Reviewer 3 Report

Ijerph-2552425, The manuscript titled "Healthcare Policies to Eliminate Neglected Tropical Diseases (NTDs) in India: A Roadmap" presents a thorough and well-structured analysis of the challenges posed by Neglected Tropical Diseases in India. The comprehensive literature review effectively highlights the current policies and programs while pinpointing their shortcomings. The proposed roadmap demonstrates a meticulous approach to addressing these gaps, offering a promising strategy for NTD elimination. This manuscript stands as a valuable contribution for policymakers and researchers working towards enhancing healthcare policies in India. But there are some highlights to improve below.   1. An abstract is a little piece of writing that concisely summarizes a manuscript's main findings, methods, and significance. It offers readers insight into the research's key contributions and innovative aspects for engagement and understanding. 2.  Please, on lines 124 to 132, avoid repeating the word "Figure 1" and revise the figure’s body as a beautiful tree. 3. The authors discussed in 3. Results and 2. Materials and Methods as being the same in both; please check these are very important parts; check figures 1 and 2. It can’t be explained the same study twice. Please strongly suggest revising them. 4. The authors mention this is a "review article" and shouldn’t use the Materials and Methods section or data in the review, or it is possible to select a different article type if applicable. Thanks, and good Luck to the authors.

Minor editing of English language required

Author Response

The manuscript titled "Healthcare Policies to Eliminate Neglected Tropical Diseases (NTDs) in India: A Roadmap" presents a thorough and well-structured analysis of the challenges posed by Neglected Tropical Diseases in India. The comprehensive literature review effectively highlights the current policies and programs while pinpointing their shortcomings. The proposed roadmap demonstrates a meticulous approach to addressing these gaps, offering a promising strategy for NTD elimination. This manuscript stands as a valuable contribution for policymakers and researchers working towards enhancing healthcare policies in India. But there are some highlights to improve below.  

Thank you for your compliment. We have tried our best to incorporate your suggestions.

1. An abstract is a little piece of writing that concisely summarizes a manuscript's main findings, methods, and significance. It offers readers insight into the research's key contributions and innovative aspects for engagement and understanding.

Thank you for your observation. We have added a sentence about the results. We hope the revision is satisfactory.

2. Please, on lines 124 to 132, avoid repeating the word "Figure 1" and revise the figure’s body as a beautiful tree.

We understand that the repetition of ‘Figure 1’ may be jarring. However, some editors prefer explicit reference to the figure. We shall defer to the final editor. I hope you agree.

About calling it a tree, we would prefer to continue to use the word ontology. We have been using the term in our earlier related publications. I hope you agree.

3. The authors discussed in 3. Results and 2. Materials and Methods as being the same in both; please check these are very important parts; check figures 1 and 2. It can’t be explained the same study twice. Please strongly suggest revising them.

We respectfully beg to differ. Figure 1 is the basic ontology, and Figure 2 is the monad map based on the ontology.  The latter visualizes and quantifies the frequency of mention of the elements in the ontology in the corpus of research studied. I hope we have clarified the difference.

4. The authors mention this is a "review article" and shouldn’t use the Materials and Methods section or data in the review, or it is possible to select a different article type if applicable. Thanks, and good Luck to the authors.

We appreciate your suggestion but would like to retain it as a review article. It is certainly not an empirical article. It proposes a theoretical (ontological) framework but goes beyond that. Given the journal’s classification, the present label probably fits the best. I hope you agree.

Round 2

Reviewer 1 Report

All the previous review recommendations have been appropriately addressed, and no further suggestions are needed.

Author Response

The reviewer says: "All the previous review recommendations have been appropriately addressed, and no further suggestions are needed." Hence, no revisions have been made in response.

Reviewer 3 Report

Thank you for giving a detailed response to the reviewer and for correcting previously reported concerns. The manuscript was accepted for publication, but the authors are still missing some correction points 2 and 3. If possible, please revise them.

Good Luck 

Author Response

We appreciate the reviewer's points 2 and 3. As we mentioned in our last response, point 2 is more an editorial choice. The editors of IJERPH appear to accept the present format.

Point 3 is one about nomenclature. The 'tree' nomenclature would be different from the one we have used in the past and would like to continue with ours. I hope the reviewer agrees.